# Establishment and Characterization of NCC-DDLPS4-C1: A Novel Patient-Derived Cell Line of Dedifferentiated Liposarcoma

**DOI:** 10.3390/jpm11111075

**Published:** 2021-10-24

**Authors:** Ryuto Tsuchiya, Yuki Yoshimatsu, Rei Noguchi, Yooksil Sin, Takuya Ono, Taro Akiyama, Takeshi Hirose, Shintaro Iwata, Akihiko Yoshida, Seiji Ohtori, Akira Kawai, Tadashi Kondo

**Affiliations:** 1Division of Rare Cancer Research, National Cancer Center Research Institute, 5-1-1 Tsukiji, Chuo-ku, Tokyo 104-0045, Japan; rytsuchi@ncc.go.jp (R.T.); yyoshima@ncc.go.jp (Y.Y.); renoguch@ncc.go.jp (R.N.); ishin@ncc.go.jp (Y.S.); takuono@ncc.go.jp (T.O.); taakiyam@ncc.go.jp (T.A.); 2Department of Orthopaedic Surgery, Graduate School of Medicine, Chiba University, 1-8-1 Inohana, Chuo-ku, Chiba 260-8670, Japan; sohtori@faculty.chib-u.jp; 3Department of Musculoskeletal Oncology, National Cancer Center Hospital, 5-1-1 Tsukiji, Chuo-ku, Tokyo 104-0045, Japan; tahirose@ncc.go.jp (T.H.); shiwata@ncc.go.jp (S.I.); akawai@ncc.go.jp (A.K.); 4Department of Diagnostic Pathology, National Cancer Center Hospital, 5-1-1 Tsukiji, Chuo-ku, Tokyo 104-0045, Japan; akyoshid@ncc.go.jp

**Keywords:** sarcoma, liposarcoma, dedifferentiated liposarcoma, patient-derived cell line

## Abstract

Dedifferentiated liposarcoma (DDLPS) is a highly malignant sarcoma characterized by the co-amplification of *MDM2* and *CDK4*. Although systemic chemotherapy is recommended for unresectable or metastatic cases, DDLPS is insensitive to conventional chemotherapy, leading to an unfavorable prognosis. Therefore, novel treatment methods are urgently required. Patient-derived cell lines are essential in preclinical studies. Recently, large-scale screening studies using a number of cell lines have been actively conducted for the development of new therapeutic drugs. However, the DDLPS cell line cannot be obtained from public cell banks owing to its rarity, hindering screening studies. As such, novel DDLPS cell lines need to be established. Accordingly, this study aimed to establish a novel DDLPS cell line from surgical specimens. The cell line was named NCC-DDLPS4-C1. NCC-DDLPS4-C1 cells retained copy number alterations corresponding to the original tumors. Further, the cells demonstrated constant growth, spheroid formation, and equivalent invasiveness to MG63 osteosarcoma cells. We also conducted drug screening and integrated the results with those of the previously reported DDLPS cell lines. Consequently, we identified the histone deacetylase inhibitor romidepsin as a novel candidate drug. In conclusion, the NCC-DDLPS4-C1 cell line is a useful tool for the basic study of DDLPS.

## 1. Introduction

Dedifferentiated liposarcoma (DDLPS) is a typical non-lipogenic sarcoma that develops from an atypical lipomatous tumor/well-differentiated liposarcoma (ALT/WDLPS) [1]. DDLPS usually affects elderly patients, and the most common site is the retroperitoneum, followed by the extremities [1,2,3,4]. Genetically, DDLPS is characterized by the amplification of *MDM2* and *CDK4* at 12q13-15 and by ALT/WDLPS [1,2,4,5,6]. In addition, multiple gene amplifications at 12q13-15, such as those of *HMGA2*, *TSPAN31*, and *CPM*, have also been identified in DDLPS [2,4,5,6,7]. Copy number alterations (CNAs) have been reported in chromosomes other than 12q13-15, and the amplification of 1p32 and 6q23 and the loss of 11q22-24 are frequently observed [8,9,10,11]. Unlike ALT/WDLPS, DDLPS has a highly aggressive potential and thus has an unfavorable prognosis.

Although complete resection is the primary modality, it is often difficult to achieve in cases involving the retroperitoneum because of the adjacent blood vessels or organs, leading to poorer prognosis than in cases arising in the extremities [3,12]. Moreover, the distant metastasis rate is as high as 15–30%, resulting in an overall mortality rate of 28% [5]. Systemic chemotherapy is considered in unresectable or metastatic cases, but DDLPS is known to be insensitive to conventional chemotherapy [13,14]. Although a recent retrospective study reported the efficacy of conventional chemotherapy on DDLPS by assessing a vascular response in addition to responsive evaluation criteria in solid tumor (RECIST), the effect on overall survival has not been well clarified [15]. Several MDM2 inhibitors have been developed in recent years, but their efficacy is yet to be investigated [16,17]. CDK4 inhibitors have also been developed and undergone clinical trials. However, the responsive rate to CDK4 inhibitors is also limited so far [18,19]. Additionally, although cancer immunotherapy has become popular in common cancers, a meta-analysis reported that DDLPS was refractory to immunotherapy [20]. Therefore, novel treatment methods are needed to improve the prognosis of DDLPS.

In vitro studies using patient-derived cell lines are essential in preclinical studies. Short-term cultured cell lines preserve the genomic features of the original tumor [21,22,23]. Recently, pharmacogenomics, the collection of genomic information, and drug screening studies using a large number of cell lines have been actively conducted for the development of new therapeutic agents [24,25,26,27]. As such, novel therapeutic agents for major cancers have been identified. However, because sarcomas are rare malignancies, the number of cell lines is extremely limited [28]. Thus, they have not benefited from pharmacogenomics [29]. In addition, there are no publicly available cell lines for DDLPS, making basic research on DDLPS difficult (Appendix A). As such, novel DDLPS cell lines need to be established.

Thus, this study aimed to establish a novel DDLPS cell line from surgical specimens. In addition to characterization of the cell line, we attempted to identify novel therapeutic candidates by integrating the results of drug screening for NCC-DDLPS1-C1 [30], NCC-DDLPS2-C1 [31], and NCC-DDLPS3-C1 [32], which we previously reported.

## 2. Materials and Methods

### 2.1. Patient History

The patient was a 77-year-old man with DDLPS. The patient visited the National Cancer Center Hospital (Tokyo, Japan) with the chief complaint of a rapidly growing mass in the left abdomen and radiating pain in the left lateral thigh. Computed tomography (CT) and needle biopsy suggested malignancy, and thus, wide resection was performed. The results indicated dedifferentiated liposarcoma. At 2 years and 2 months postoperative, CT detected local recurrence in the left retroperitoneum (Figure 1a). Given that there was no evidence of distant metastasis, the patient underwent re-operation. Histologically, the tumor mostly consisted of a dedifferentiated component showing dense proliferation of pleomorphic spindle cells (Figure 1b). A well-differentiated liposarcoma component was present at the tumor periphery (Figure 1c). The tumor harbored high-level *MDM2* amplification using fluorescence in situ hybridization (FISH) (Figure 1d). A year after the reoperation, re-recurrence was observed. Eventually, the patient died 4 years after the initial surgery.

A part of the resected tumor at re-operation was used to establish the cell line described in this study. The cell line was named NCC-DDLPS4-C1.

### 2.2. Pathological Examination

Pathological examination was performed on 4 μm-thick sections from a paraffin-embedded tumor specimen. The deparaffinized sections were stained with hematoxylin and eosin (H&E) and assessed by an expert pathologist.

### 2.3. FISH Analysis

FISH analysis was also performed on 4 µm-thick sections of the tumor specimen. *MDM2* amplification was detected using the ZytoLight SPEC MDM2/CEN 12 Dual Color Probe (ZytoVison GmgH, Bremerhaven, Germany). The hybridized images were obtained using the Metafer Slide Scanning Platform (MetaSystems, Altlussheim, Germany).

### 2.4. Primary Cell Culture Procedure

The surgically resected tumor was first stored in CELLBANKER (Nippon Zenyaku Kogyo Co., Ltd., Fukushima, Japan) at −80 °C. The tumor was stored for 1 year and 11 months before culturing. At the time of primary culture, the cryopreserved tumor specimen was thawed and divided into small pieces with sterile scissors. The tumor specimens were cultured in DMEM/F12 (Gibco, Grand Island, NY, USA) supplemented with 5% heat-inactivated fetal bovine serum (Gibco), 100 μg/mL penicillin and 100 μg/mL streptomycin (Nacalai Tesque, Kyoto, Japan), 0.4 µg/mL hydrocortisone (Sigma-Aldrich, St. Louis, MO, USA), 5 ng/mL EGF (Sigma-Aldrich), 10 ng/mL bFGF (Sigma-Aldrich), 5 µg/mL insulin (Sigma-Aldrich), and 10 µM Y-27632 (Selleck Chemicals, Houston, TX, USA) at 37 °C in a humidified atmosphere with 5% CO_2_. The cells that developed from the tumor specimens were maintained for more than 6 months under tissue culture conditions and were passaged more than 25 times.

### 2.5. Authentication and Quality Control

Short tandem repeats (STRs) of the established cell line were examined at 10 loci using the GenePrint 10 system (Promega, Madison, WI, USA). Briefly, genomic DNA was extracted from the established cell line and the corresponding tumor tissue using AllPrep DNA/RNA Mini kits (Qiagen, Hilden, Germany). STRs were amplified using the obtained DNA (10 ng). PCR products were analyzed using a 3500xL Genetic Analyzer (Applied Biosystems, Foster City, CA, USA). The obtained data were analyzed using the GeneMapper software (Applied Biosystems). Finally, the STR profiles were examined using a cell line database, Cellosaurus [33], to check whether they matched with any other cell lines.

Mycoplasma contamination was examined using the e-Myco Mycoplasma PCR Detection Kit (Intron Biotechnology, Suwon, Korea). Extracted cellular DNA (50 ng) was used for PCR. The DNA sequence unique to mycoplasma was amplified. The amplified DNA was electrophoresed, and the gel was stained with Midori Green Advance stain (Nippon Genetics, Tokyo, Japan). Finally, the images were obtained using the Amersham Imager 600 system (GE Healthcare, Little Chalfont, UK).

### 2.6. Single-Nucleotide Polymorphism Array Analysis

Single nucleotide polymorphism (SNP) array analysis was performed using the Infinium OmniExpressExome-8 v1.4 BeadChip (Illumina, San Diego, CA, USA). Briefly, the extracted genomic DNA of the established cell line and the corresponding tumor tissue were amplified and hybridized onto array slides in an iScan system (Illumina). Log R ratios were calculated using the Genome Studio 2011.1 (Illumina) and analyzed using the R version 4.0.3 (R Foundation for Statistical Computing, http://www.R-project.org, accessed on 26 September 2021) and DNAcopy package version 1.64.0 (Bioconductor, https://bioconductor.org/, accessed on 26 September 2021). Chromosome regions with copy numbers >3 or <1 were regarded as amplifications and losses, respectively. Genes that exhibited CNAs were annotated using the biomaRt package version 2.46.0 (Bioconductor) and the GRCh 37 assembly in Ensembl (https://asia.ensembl.org/index.html, accessed on 26 September 2021).

### 2.7. Western Blotting

Western blotting was conducted to determine the expression levels of MDM2 and CDK4 using the established cell line, with the WDLPS cell line (93T449; ATCC, Manassas, VA, USA) as a positive control and normal human foreskin fibroblast (HFF) cell line (BioWhittaker, Basel, Switzerland) as a negative control according to a previously reported procedure [30]. First, antibodies against MDM2 (1:100; Calbiochem, San Diego, CA, USA), CDK4 (1:1000; Santa Cruz Biotechnology, Dallas, TX, USA), and β-actin (1:5000; Abcam, Cambridge, UK) were used with their corresponding secondary antibodies (ECL anti-mouse IgG and ECL anti-rabbit IgG; 1:5000; GE Healthcare). Images were obtained using an Amersham Imager 600 (GE Healthcare) after treatment with Western Lightning Plus-ECL (PerkinElmer, Waltham, MA, USA). The relative expression levels of each protein were compensated for by β-actin.

### 2.8. Cell Proliferation Assay

Cell proliferation assays were performed using the Cell Counting Kit-8 (Dojindo Laboratories, Kumamoto, Japan). The cells (1 × 10^4^) were seeded in 24-well culture plates on day 0. The number of cells was assessed based on the absorbance at 450 nm after treatment with the Cell Counting Kit-8. Absorbance was measured every other day until day 4. The doubling time was calculated based on the growth curve obtained.

### 2.9. Spheroid Formation Assay

Spheroid formation was assessed using 96-well round-bottom ultra-low attachment microplates (Corning, Inc., Corning, NY, USA). Cells (1 × 10^5^) were seeded in a plate and incubated for 3 days. Spherical colonies were embedded in paraffin and fixed with iPGell (Genostaff, Tokyo, Japan). Then, paraffin sections were prepared for H&E staining.

### 2.10. Invasion Assay by Real-Time Cell Analyzer

Invasion assays were performed using a real-time cell analyzer (xCELLigence, Agilent, Santa Clara, CA, USA). Briefly, the cells (4 × 10^4^) suspended in serum-free DMEM/F12 medium were seeded in the upper chamber coated with BD Matrigel matrix (BD Biosciences, Franklin Lakes, NJ, USA). The culture medium, which was used to maintain the cells, was added to the lower chamber. Then, the electronic sensors detected the cells that migrated from the upper chamber to the lower chamber. Measurements were taken every 15 min for 72 h. In this experiment, MG63 osteosarcoma cells (JCRB; Ibaraki, Osaka, Japan) were used as controls [34].

### 2.11. Tumorigenesis Assessment in Nude Mice

Animal experiments were conducted following the guidelines of the Institute for Laboratory Animal Research, National Cancer Center Research Institute. Briefly, the cells (1 × 10^6^) were suspended in a 100 μL 1:1 mixture of BD Matrigel matrix and D-PBS (-) (Nakalai Tesque, Inc., Kyoto, Japan) and injected subcutaneously into four locations of female BALB/c nude mice (CLEA Japan, Inc., Tokyo, Japan). Tumor size was measured weekly, and the volume was estimated according to the following formula: volume = (length × width2)/2. After 2 months, the tumors were surgically excised and prepared for H&E staining.

### 2.12. Drug Screening Test

Drug screening tests were performed using 197 drugs, including FDA-approved anticancer agents (Selleck Chemicals; Appendix A). In this experiment, the cells and reagents were disseminated using a Bravo automated liquid handling platform (Agilent Technologies, Santa Clara, CA, USA). Briefly, the cells (1 × 10^4^) were seeded in a 384-well plate and incubated for 1 d. On the second day, 10 µM of each drug was added. On the fifth day, cell proliferation was measured based on the absorbance at 450 nm using a Cell Counting Kit-8. Cell viability was calculated by comparing with the DMSO control. The obtained data were integrated with the results of NCC-DDLPS1-C1 [30], NCC-DDLPS2-C1 [31], and NCC-DDLPS3-C1 [32], which we previously reported, and quantile normalized using R version 4.0.3, limma package version 3.46.0 (Bioconductor). Unsupervised hierarchical clustering was performed using gplots package version 3.1.0 (CRAN, https://cran.r-project.org, accessed on 26 September 2021).

Subsequently, the half-maximal inhibitory concentration (IC_50_) value was calculated for the drugs that exhibited remarkable antiproliferative effects in drug screening and standard treatment drugs. In short, the selected drugs were dispensed in the cells seeded as the preceding procedure at 10 different concentrations from 0.1–100,000 nM. Based on the cell viabilities calculated in the same way, IC_50_ values were determined using GraphPad Prism 9.1.1 (GraphPad Software, San Diego, CA, USA).

## 3. Results

### 3.1. Authentication of the Established Cell Line

The NCC-DDLPS4-C1 cell line was authenticated by analyzing the STR at 10 loci (Table 1, Appendix A). The STR match ratio between the cells and the original tumor was 93.3%, as calculated by the Tanabe formula [35]. This score met the 80% threshold [36], guaranteeing that the NCC-DDLPS4-C1 cell line was established from the original tumor. Furthermore, none of the other cell lines had an STR match ratio of over 80%, according to the Cellosaurus. Therefore, NCC-DDLPS4-C1 was not contaminated with other cell lines or a novel cell line. Moreover, the DNA sequence unique to mycoplasma was not detected in the cellular DNA, confirming that the NCC-DDLPS4-C1 cell line was not contaminated with mycoplasma (data not shown).

### 3.2. Genomic Landscape of the Cell Line

SNP array analysis revealed that NCC-DDLPS4-C1 cells harbored multiple CNAs (amplification: 11p14, 12q13-15, 12q21-23, and 15q22. Losses: 3q29, 8p23, 11q23, 15q21, 18q21, and 19q13; Figure 2), corresponding to the original tumor tissue. The representative CNAs are summarized in Table 2, and all the CNAs are listed in Appendix A. Among the genes with CNAs, *MDM2* and *CDK4* were co-amplified, which is characteristic of DDLPS. In addition to *MDM2* and *CDK4*, multiple genes with CNAs were observed.

### 3.3. MDM2 and CDK4 Expressions of the Cell Line

In addition to the amplification of *MDM2* and *CDK4* by SNP array, we confirmed the high expression of *MDM2* and *CDK4* by Western blotting (Figure 3a). These findings also support that the NCC-DDLPS4-C1 cell line retains the characteristics of the original tumor. The relative expression of each protein compensated for by β-actin is shown in Figure 3b,c.

### 3.4. In Vitro Characteristics of the Cell Line

Microscopically, NCC-DDLPS4-C1 cells comprised elongated spindle cells under culture conditions (Figure 4a,b). Using the cell proliferation assay, we obtained the growth curve of NCC-DDLPS4-C1 cells (Figure 4c). The growth curve indicated that NCC-DDLPS4-C1 cells proliferated steadily, and the population doubling time of NCC-DDLPS4-C1 cells was 72.5 h. The results of the spheroid formation assay proved that NCC-DDLPS4-C1 cells were capable of spheroid formation. Dense proliferation of pleomorphic spindle cells with nuclear atypia was observed in the H&E-stained spheroid section (Figure 4d). In the invasion assay, the invasiveness of NCC-DDLPS4-C1 cells was comparable to that of MG63 cells (Figure 4e).

### 3.5. Tumorigenesis in Nude Mice

Among the four locations in which NCC-DDLPS4-C1 cells were injected in BALB/c nude mice, tumor growth was observed in one location (Appendix A). The tumor comprised dense proliferation of small oval cells (Appendix A). However, tumors did not develop sufficiently in the other three locations, and the estimated tumor volumes did not increase consistently (Appendix A).

### 3.6. Sensitivity to Anticancer Drugs

The antiproliferative effects of 197 drugs on NCC-DDLPS4-C1 cells were examined. The cell viability after treatment with each drug at a concentration of 10 µM is shown in Appendix A. The antiproliferative effects of each drug on NCC-DDLPS1-C1 [30], NCC-DDLPS2-C1 [31], and NCC-DDLPS3-C1 [32], which we previously reported, are also described. Unsupervised hierarchical clustering revealed that the drugs were categorized into three clusters (cluster A: effective group; cluster B: intermediate effect group; cluster C: poor effect group) according to each antiproliferative effect (Figure 5a). Cluster A contained a high proportion of cytotoxic agents, followed by tyrosine kinase inhibitors and molecular-targeted agents (Figure 5b). As for cytotoxic agents, cluster A included a high proportion of topoisomerase inhibitors, while alkylating agents and antimetabolite agents did not show sufficient antiproliferative effects (Figure 5c). Regarding tyrosine kinase inhibitors, anaplastic lymphoma kinase (ALK) inhibitors showed promising antiproliferative effects, although the number of ALK inhibitors used in this experiment was small (Figure 5d). Histone deacetylase (HDAC) inhibitors and MDM2 inhibitors, which belong to molecular-targeted agents, were categorized into only clusters A and B (Figure 5e).

The IC_50_ values of NCC-DDLPS4-C1 were determined for the drugs that were used in the screening study. The IC_50_ values of NCC-DDLPS4-C1, NCC-DDLPS1-C1, NCC-DDLPS2-C1, and NCC-DDLPS3-C1, which we previously reported, are listed in Appendix A. Notably, the HDAC inhibitor romidepsin, doxorubicin, and trabectedin demonstrated remarkable antiproliferative effects. These drugs are used as standard therapies for sarcomas (Figure 6, Table 3).

## 4. Discussion

DDLPS is a highly malignant sarcoma with an unfavorable prognosis. Although systemic chemotherapy is recommended for unresectable or metastatic cases, conventional chemotherapy has no overall survival benefit [13]. Preclinical studies using patient-derived cell lines are essential for the development of novel treatment methods. However, the number of DDLPS cell lines reported to date is small and it is not deposited in public cell banks owing to the rarity of DDLPS. This hinders the development of new drugs. In this study, we established a novel DDLPS cell line named NCC-DDLPS4-C1 from surgical specimens of DDLPS patients and performed drug screening with detailed characterization.

We established the NCC-DDLPS4-C1 cell line from the recurrent tumor of elderly patients with DDLPS. In addition to the typical elderly age of patients, another common feature of DDLPS is retroperitoneal origin. Thus, the NCC-DDLPS4-C1 cell line established from an elderly DDLPS patient was useful for understanding the typical nature of DDLPS.

The NCC-DDLPS4-C1 cell line harbored CNAs similar to those of the original tumor, indicating that the NCC-DDLPS4-C1 cell line retained its genomic characteristics under the culturing conditions. In addition to the co-amplification of *MDM2* and *CDK4*, multiple gene amplifications were observed in chromosome 12q13-15 of the NCC-DDLPS4-C1 cell line. The co-amplification of *HMGA2*, *TSPAN31*, and *YEATS4* with *MDM2* is frequently observed in DDLPS and is involved in tumor development [7,37]. *CPM* knockdown resulted in inhibition of DDLPS cell proliferation, migration, and invasion, suggesting an association with malignant behavior [38]. Simultaneous amplification of *SLC35E3*, *MDM2*, and *CPM* is also involved in the development of DDLPS [11]. The amplification of *FRS2* is also frequently observed in DDLPS. *FRS2* is related to the FGF receptor signaling pathway, suggesting that it may be a therapeutic target [39,40]. Except for the amplification of chromosome 12q13-15, the NCC-DDLPS4-C1 cell line exhibited a loss of 11q23, which was not observed in NCC-DDLPS1-C1 [30], NCC-DDLPS2-C1 [31], and NCC-DDLPS3-C1 [32]. Loss of 11q23-24 was observed in 44% of DDLPS cases and was associated with increased genomic complexity [10]. However, the function of this loss remains unclear and requires further investigation.

NCC-DDLPS4-C1 cells demonstrated typical spindle cell morphology under culture conditions and the capability of spheroid formation. The co-expression of MDM2 and CDK4 also supported that the NCC-DDLPS4-C1 cell line maintained the characteristics of DDLPS. Compared to the DDLPS cell lines we previously reported [30,31,32], the NCC-DDLPS4-C1 cell line demonstrated slower proliferation and less invasiveness. This relatively moderate in vitro characteristic may be due to multiple genetic factors, but the detailed mechanism is still unclear. Contrary to the in vitro usability, in vivo tumorigenesis was insufficient under the described conditions. The difference of the tumor microenvironment between humans and mice, and the site of origin may affect the tumorigenicity in vivo. In addition, the degree of immunodeficiency of mice may also affect the tumorigenicity. Thus, the NCC-DDLPS4-C1 cell line may not be suitable for in vivo experiments using nude mice.

Drug screening identified multiple novel therapeutic candidates. Among them, the HDAC inhibitor romidepsin showed equivalent to superior antiproliferative effects on NCC-DDLPS4-C1 cells, along with NCC-DDLPS1-C1 [30], NCC-DDLPS2-C1 [31], and NCC-DDLPS3-C1 [32] cells, compared with the standard treatment drugs doxorubicin and trabectedin. The other research group also reported the efficacy of HDAC inhibitor on liposarcoma cell lines by medium-scale high-throughput drug screening [41]. A recent study reported that the HDAC gene family was amplified in sarcomas, including in DDLPS, and that the combination of HDAC inhibitor and anti-PD1 therapy promoted tumor regression [42]. Another study suggested that HDAC inhibition reduced MDM2 expression and tumor growth in DDLPS [43]. However, the number of studies investigating the efficacy of HDAC inhibitors for DDLPS is still limited, and further studies are needed.

A strength of this study is that a novel DDLPS cell line, NCC-DDLPS4-C1, was established. Further, novel therapeutic candidates, such as romidepsin, were identified. However, this study also has some limitations. Although this study is one of the largest studies to summarize the results of drug screening for DDLPS, the number of cell lines is still lower than in other common cancers. For example, it was reported that 30 cell lines are needed in a drug screening test to detect novel therapeutic drugs at a statistical power of 70% [44]. In addition, the relationship between genomic information and drug sensitivity could not be identified because of the small number of DDLPS cell lines. Therefore, continuous efforts to establish novel DDLPS cell lines are required.

## 5. Conclusions

We successfully established a novel DDLPS cell line, NCC-DDLPS4-C1. NCC-DDLPS4-C1 cells retained the characteristics of the original tumors. We also found that the HDAC inhibitor romidepsin has a remarkable antiproliferative effect on NCC-DDLPS4-C1 cells. However, the number of DDLPS cell lines remains insufficient. More DDLPS cell lines need to be established to reveal the complexity and diversity of DDLPS. The established cell line should be broadly shared in the research community to promote the study of rare malignancies such as DDLPS.

## Figures and Tables

**Figure 1 jpm-11-01075-f001:**
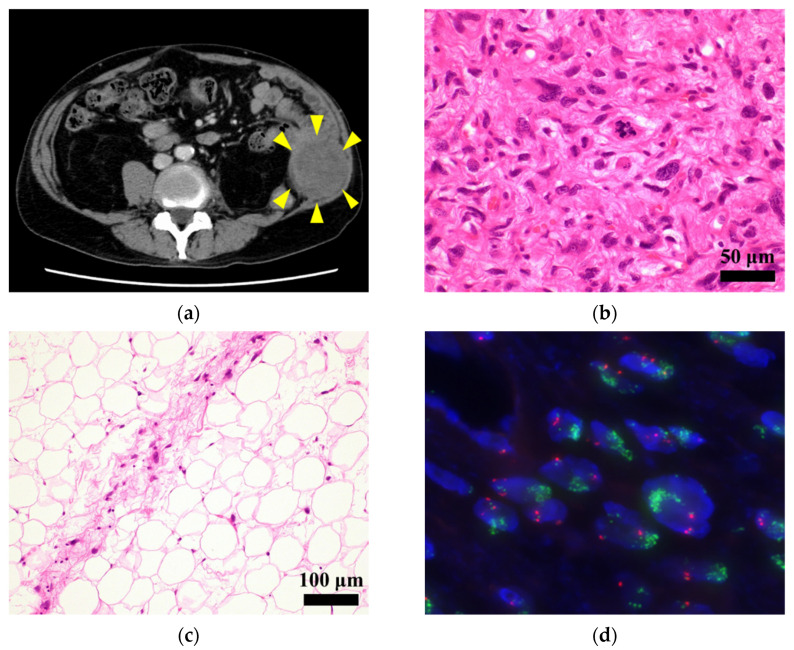
Clinicopathological data. (**a**) Computed tomography showing a recurrent tumor in the left retroperitoneum. Yellow arrows indicate the tumor; (**b**) the tumor consisting of pleomorphic spindle cells in hematoxylin and eosin stained section; (**c**) well-differentiated component observed at the rumor periphery; (**d**) fluorescence in situ hybridization (FISH) analysis showing high-level MDM2 amplification (green signals).

**Figure 2 jpm-11-01075-f002:**
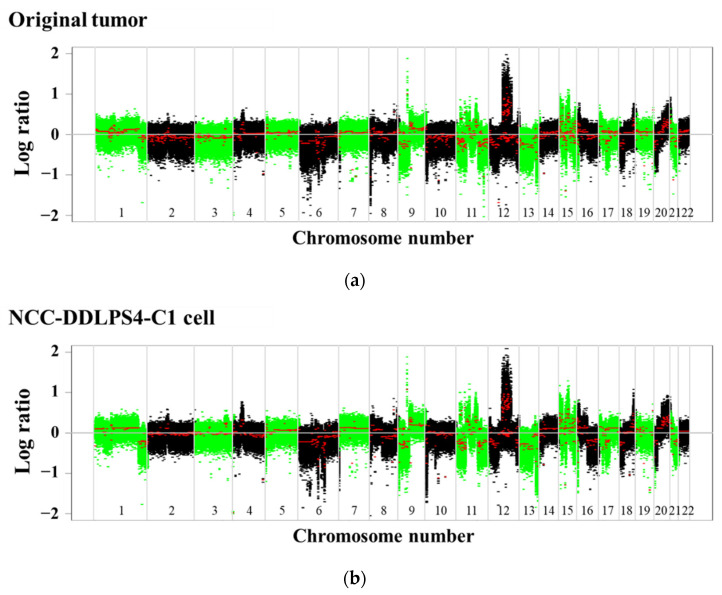
Single-nucleotide polymorphism (SNP) array analysis. Copy number alterations in (**a**) the original tumor and (**b**) NCC-DDLPS4-C1 cells were demonstrated. The *X*-axis indicates the log ratio of copy number. *Y*-axis indicates chromosomal location. The characteristic amplification of chromosome 12 was observed.

**Figure 3 jpm-11-01075-f003:**
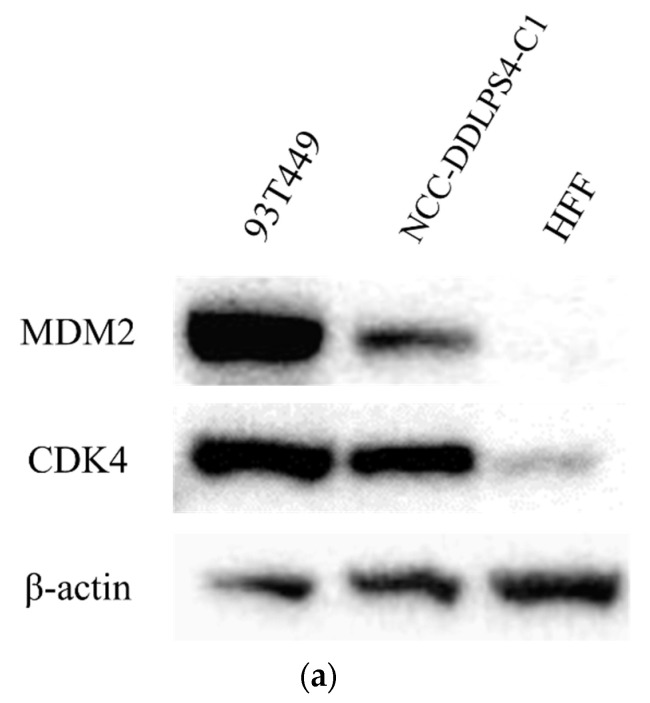
Western blot analysis. (**a**) NCC-DDLPS4-C1 cells with MDM2 and CDK4 overexpression; (**b**,**c**) relative expressions of MDM2 and CDK4, respectively.

**Figure 4 jpm-11-01075-f004:**
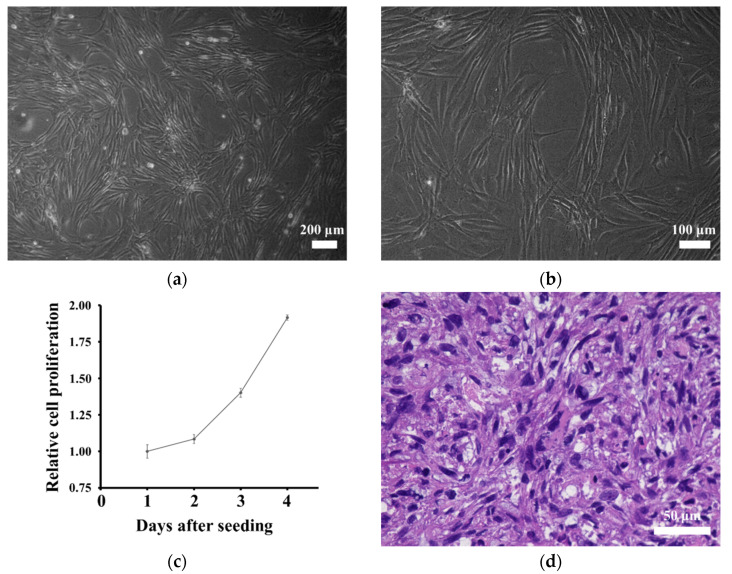
In vitro features of NCC-DDLPS4-C1 cells. (**a**,**b**) NCC-DDLPS4-C1 cells showing elongated spindle cell under culturing conditions; (**c**) growth curve of NCC-DDLPS4-C1 cells. The *X*-axis and *Y*-axis indicate the relative cell proliferation of NCC-DDLPS4-C1 cell and the day after seeding, respectively; (**d**) the spheroid consisting of pleomorphic spindle cells with nuclear atypia in hematoxylin and eosin stained section; (**e**) the invasion ability of NCC-DDLPS4-C1 cells compared to that of MG63 osteosarcoma cells.

**Figure 5 jpm-11-01075-f005:**
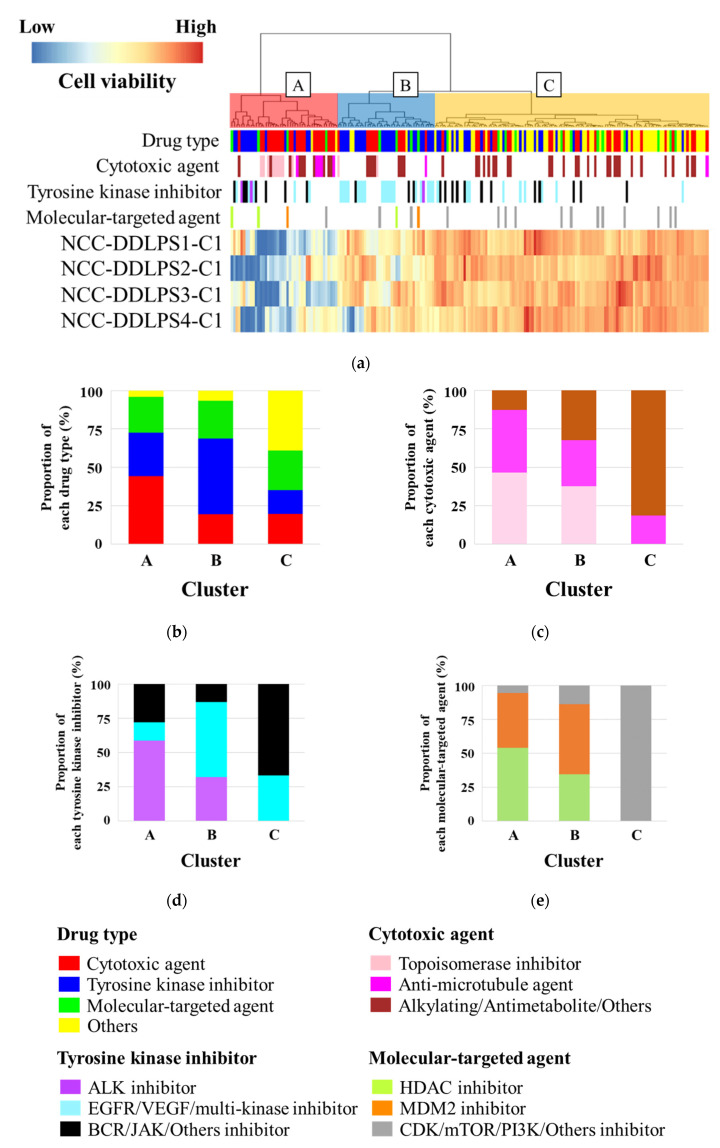
Drug screening using 197 drugs. (**a**) Unsupervised hierarchical clustering showing the antiproliferative effect of each drug on NCC-DDLPS4-C1 cells. Drugs are categorized into three clusters (cluster A: effective group; cluster B: intermediate effect group; cluster C: poor effect group). (**b**–**e**) The proportion of each drug type categorized into clusters A, B, and C. The graphs are depicted after normalization of the number of drugs.

**Figure 6 jpm-11-01075-f006:**
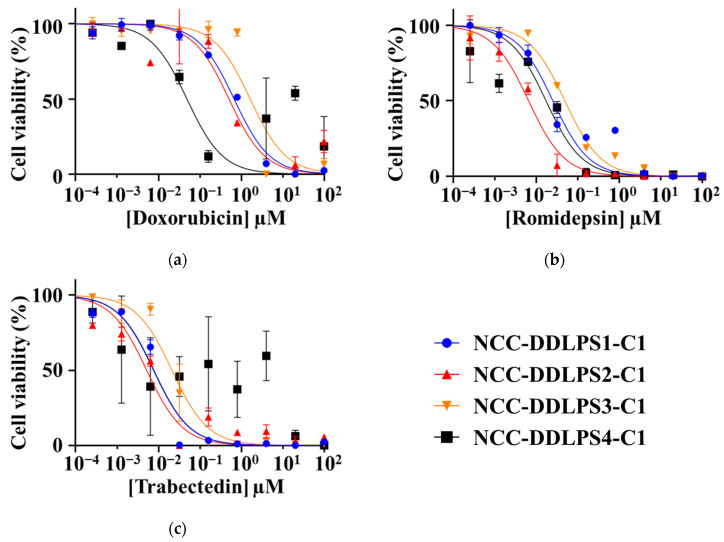
Growth curves for the IC_50_ value calculation. Cell viability after the treatment with (**a**) doxorubicin, (**b**) romidepsin, and (**c**) trabectedin at different concentrations is described. The data of NCC-DDLPS1-C1, NCC-DDLPS2-C1, and NCC-DDLPS3-C1 are previously published [31,32].

**Table 1 jpm-11-01075-t001:** Short tandem repeat analysis.

Microsatellite	NCC-DDLPS4-C1	Original Tumor Tissue
(Chromosome)		
Amelogenin (X Y)	X, Y	X, Y
TH01 (3)	7	7, 9
D21S11 (21)	29, 30	29, 30
D5S818 (5)	11, 12	11, 12
D13S317 (13)	12	12
D7S820 (7)	8, 11	8, 11
D16S539 (16)	9	9
CSF1PO (5)	10, 12	10, 12
vWA (12)	17	14, 17
TPOX (2)	8, 11	8, 11

**Table 2 jpm-11-01075-t002:** Representative copy number alterations.

Gene Symbol	Chromosome Region	Copy Number	Type
ATF4P4	11q23.2	0.1	Loss
LRRC37A13P	11q23.2	0.1	Loss
USP28	11q23.2	0.1	Loss
TSPAN31	12q14.1	3.0	Amp
CDK4	12q14.1	3.0	Amp
HMGA2	12q14.3	3.4	Amp
SLC35E3	12q15	3.5	Amp
MDM2	12q15	3.5	Amp
CPM	12q15	3.5	Amp
YEATS4	12q15	3.4	Amp
FRS2	12q15	3.4	Amp

Amp: amplification.

**Table 3 jpm-11-01075-t003:** Summary of half-maximal inhibitory concentration (IC_50_) values (µM).

CAS#	Drug	NCC-DDLPS1-C1	NCC-DDLPS2-C1	NCC-DDLPS3-C1	NCC-DDLPS4-C1
25316-40-9	Doxorubicin	0.6706	0.53	1.642	0.04878
128517-07-7	Romidepsin	0.02538	0.0068	0.0493	0.01734
114899-77-3	Trabectedin	0.007564	0.0049	0.0212	0.007721

The data of NCC-DDLPS1-C1, NCC-DDLPS2-C1, and NCC-DDLPS3-C1 are previously published [31,32].

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
