# Peer review of "Establishment and Characterization of NCC-DDLPS4-C1: A Novel Patient-Derived Cell Line of Dedifferentiated Liposarcoma"

_jpm, 2021, doi:10.3390/jpm11111075_

Round 1

Reviewer 1 Report

The authors Ryuto Tsuchiya et al present work on development of a new dedifferentiated liposarcoma cell line derived from relapsed sarcoma in the retroperitoneum of an elderly man. The line was derived from tumor that had been frozen shortly after resection. The authors carefully characterized the cell line and present sound evidence that the line genomically resembles the liposarcoma that was resected. While the line may be valuable for in vitro testing, it did not grow well in a PDX mouse model and may have limited usefulness in mouse in vivo studies. In the drug screen methodology used, the authors found the novel cell line and previously described dedifferentiated liposarcoma cell lines were sensitive to exposure to anthracyclines, particularly doxorubicin, trabectedin and the HDAC inhibitor romidepsin. The cell lines examined were not sensitive to the CDK4 inhibitors that were tested and demonstrated variable and intermediate sensitivity to the MDM2 inhibitor (RG-7112) tested. This novel cell line may beneficial to the research community to help dissect pathways critical to maintenance of cell viability and survival and preclinical testing of novel agents, but impact will be limited if it is unable to establish growth in animal models.

I have a few minor comments that may improve the manuscript.

The authors discuss in the introduction that dedifferentiated liposarcoma is insensitive to conventional liposarcoma. This statement ignores important data presented in Livingston, J.A., Bugano, D., Barbo, A. et al. Role of chemotherapy in dedifferentiated liposarcoma of the retroperitoneum: defining the benefit and challenges of the standard. Sci Rep 7, 11836 (2017). https://doi.org/10.1038/s41598-017-12132-w that demonstrates activity of doxorubicin and ifosfamide and also ignores data presented in the in vitro work that demonstrates tumor response to doxorubicin. The work presented supports use of doxorubicin as an active agent in dedifferentiated liposarcoma.

The authors do not present background data on clinical activity of CDK4 inhibitors in the introduction which is a stark omission. I refer the authors to Dickson MA, Schwartz GK, Keohan ML, et al. Progression-Free Survival Among Patients With Well-Differentiated or Dedifferentiated Liposarcoma Treated With CDK4 Inhibitor PalbociclibA Phase 2 Clinical TrialJAMA Oncol. 2016;2(7):937–940. doi:10.1001/jamaoncol.2016.0264 and DOI: 10.1200/JCO.2019.37.15_suppl.11004 Journal of Clinical Oncology 37, no. 15_suppl (May 20, 2019) 11004-11004. While the authors in vitro research suggests CDK4 inhibitors are not active against the cell lines tested, clinical development of CDK4 inhibitors is underway - SARC041: Study of Abemaciclib Versus Placebo in Patients With Advanced Dedifferentiated Liposarcoma - ClinicalTrials.gov Identifier: NCT04967521.

In the cell viability assay, the results to doxorubicin exposure in cell line DDLPS4-C1 seem much more variable than the 3 other cell lines, and viability to 10-1 concentration seems to be an outlier. Do the authors have an explanation for the variability or have they repeated the experiment to see if the growth curve of DDLPS4-C1 to doxorubicin is reproducible?

For how long was the resected liposarcoma frozen at -80o before it was thawed and cultured?

Author Response

Reviewer #1

We are grateful to Reviewer #1 for the critical and constructive comments.

Comment 1

The authors discuss in the introduction that dedifferentiated liposarcoma is insensitive to conventional liposarcoma. This statement ignores important data presented in Livingston, J.A., Bugano, D., Barbo, A. et al. Role of chemotherapy in dedifferentiated liposarcoma of the retroperitoneum: defining the benefit and challenges of the standard. Sci Rep 7, 11836 (2017). https://doi.org/10.1038/s41598-017-12132-w that demonstrates activity of doxorubicin and ifosfamide and also ignores data presented in the in vitro work that demonstrates tumor response to doxorubicin. The work presented supports use of doxorubicin as an active agent in dedifferentiated liposarcoma.

Response to Comment 1

Thank you for providing an interesting research about the efficacy of conventional chemotherapy on dedifferentiated liposarcoma. Accordingly, we mentioned the article in the introduction section.

Comment 2

The authors do not present background data on clinical activity of CDK4 inhibitors in the introduction which is a stark omission. I refer the authors to Dickson MA, Schwartz GK, Keohan ML, et al. Progression-Free Survival Among Patients With Well-Differentiated or Dedifferentiated Liposarcoma Treated With CDK4 Inhibitor Palbociclib: A Phase 2 Clinical Trial. JAMA Oncol. 2016; 2(7):937–940. doi:10.1001/jamaoncol.2016.0264 and DOI: 10.1200/JCO.2019.37.15_suppl.11004 Journal of Clinical Oncology 37, no. 15_suppl (May 20, 2019) 11004-11004. While the authors in vitro research suggests CDK4 inhibitors are not active against the cell lines tested, clinical development of CDK4 inhibitors is underway - SARC041: Study of Abemaciclib Versus Placebo in Patients With Advanced Dedifferentiated Liposarcoma - ClinicalTrials.gov Identifier: NCT04967521.

Response to Comment 2

Accordingly, we mentioned CDK4 inhibitors in the introduction section and added the references.

Comment 3

In the cell viability assay, the results to doxorubicin exposure in cell line DDLPS4-C1 seem much more variable than the 3 other cell lines, and viability to 10-1 concentration seems to be an outlier. Do the authors have an explanation for the variability or have they repeated the experiment to see if the growth curve of DDLPS4-C1 to doxorubicin is reproducible?

Response to Comment 3

We performed this experiment duplicates. Although there is a possibility of experimental errors, we evaluate the variation of cell seeding and drug addition by calculating coefficient of variation (CV) value. We generally conduct this experiment so that the CV value is below 15%, and this criteria does not deviate from previous studies. In addition, there is no definitive standard CV value criteria in drug screening. Furthermore, the outlier does not essentially affect the utility of this cell line and discussion. Therefore, we consider our data to be acceptable.

Comment 4

For how long was the resected liposarcoma frozen at -80°C before it was thawed and cultured?

Response to Comment 4

We stored the tumor specimens at -80°C for 1 year and 11 months, and described it in the revised manuscript.

Reviewer 2 Report

In this manuscript the authors present the establishment and characterization of a novel cell line derived from a patient with dedifferentiated liposarcoma. The authors have tested the capacity to form spheroids, the invasiveness and the in vivo tumor-formation capacity of the cell line. They also performed drug screening on a panel of 197 drugs. Indeed, patient-derived cell lines constitute major tools in preclinical studies and DDLPS cell lines are not available from public cell banks. There is therefore an urgent need for establishment and validation of cell lines derived from DDLPS primary tumors. As such, this manuscript provides an interesting contribution to the field of research on liposarcoma.

However, the authors should consider and respond to the following points and comments before consideration for publication in Journal of personalized medicine.

  • On which criteria were chosen the « representative copy number alterations » in Table 2 ?

The FRS2 gene in the 12q13-15 amplicon is not mentioned although its amplification has been demonstrated in the vast majority of DDLPS and several publications have recently shown the potential relevance of pan-FGFR inhibitors such as erdafitinib and LY2874455 as candidates for anti-liposarcoma therapy (Hanes R et al, Cells 2019, 8, 189 ; Dadone-Montaudie B et al, Cancers 2020, 12, 3058)

  • The legends are missing for the Supplementary figures, only the titles are mentioned, the paragraph on the in vivo experiments (lines 255-260 p9) could be enriched even if the results are negative. Since tumor-forming capacity in vivo is a key property for a cancer cell line, the authors might provide hypothesis on the characteristics of this cell line which may explain

Its lack of tumorigenicity in vivo.

  • Regarding the spheroid formation assay, have the authors tried to grow spheroids of this cell line embedded in matrigel to test their capacity of invading an extracellular matrix ?
  • It is very interesting to compare the differential responses of the 4 cell lines established by the authors to the panel of drugs they have tested. But the drug response curves (Figure 6) for trabectedin and doxorubicin for the DDLPS4 cell line need improvement.

In table S5, when the IC50 values are missing for a cell line is it because the cell line did not respond to the first 10 uM drug concentration and the dose response experiment has not been performed ?

The authors should improve the discussion comparing the results of their screen with the results obtained by other teams on drug screening on liposarcoma, for example Grad and Coll 2021 PLoS ONE 16(3) : e0248140

Author Response

Reviewer #2

We truly appreciate the constructive comments by Reviewer #2.

Comment 1

On which criteria were chosen the « representative copy number alterations » in Table 2?

Response to Comment 1

We listed the genes that have been reported to relate the malignant potential and pathogenesis of DDLPS in the previous studies.

Comment 2

The FRS2 gene in the 12q13-15 amplicon is not mentioned although its amplification has been demonstrated in the vast majority of DDLPS and several publications have recently shown the potential relevance of pan-FGFR inhibitors such as erdafitinib and LY2874455 as candidates for anti-liposarcoma therapy (Hanes R et al, Cells 2019, 8, 189; Dadone-Montaudie B et al, Cancers 2020, 12, 3058)

Response to Comment 2

Thank you for kindly introducing the interesting research. We discussed about it and added the references. We also added FRS2 gene in Table 2.

Comment 3

The legends are missing for the Supplementary figures, only the titles are mentioned, the paragraph on the in vivo experiments (lines 255-260 p9) could be enriched even if the results are negative. Since tumor-forming capacity in vivo is a key property for a cancer cell line, the authors might provide hypothesis on the characteristics of this cell line which may explain its lack of tumorigenicity in vivo.

Response to Comment 3

Accordingly, we described the legend of supplementary figures.

Although the tumorigenicity in vivo was considered a key property for a cancer cell lines, there are various cell lines without tumor formation capacity based on our experience. The difference of the tumor microenvironment between human and mice, and the site of origin may affect the tumorigenicity in vivo. In addition, the degree of immunodeficiency of mice may also affect tumor formation. However, since there are a number of immunodeficiency mice, it is difficult to assess evaluate tumorigenicity in all of them. Therefore, we only evaluate tumorigenicity in the representative immunodeficiency mice, nude mice. We described this content in the discussion section.

Comment 4

Regarding the spheroid formation assay, have the authors tried to grow spheroids of this cell line embedded in matrigel to test their capacity of invading an extracellular matrix?

Response to Comment 4

We only assessed the capability of spheroid formation, and did not evaluate the invasiveness of the spheroid. The invasiveness of spheroids is very interesting, but this study is not focused on the characteristics of spheroids. Therefore, we would like to evaluate the spheroid characteristics in a future research.

Comment 5

It is very interesting to compare the differential responses of the 4 cell lines established by the authors to the panel of drugs they have tested. But the drug response curves (Figure 6) for trabectedin and doxorubicin for the DDLPS4 cell line need improvement.

Response to Comment 5

In this drug screening test, we evaluate the variation of cell seeding and drug addition by calculating coefficient of variation (CV) value, which is often assessed in similar experiments. We generally conduct this experiment so that the CV value is below 15%, and this criteria does not deviate from previous studies. In addition, there is no definitive standard CV value criteria in drug screening. Therefore, we consider our data to be acceptable.

Comment 6

In table S5, when the IC50 values are missing for a cell line is it because the cell line did not respond to the first 10 uM drug concentration and the dose response experiment has not been performed?

Response to Comment 6

Yes, exactly. The IC50 values were calculated only for the drugs that showed antitumor effects at the first 10 μM concentration.

Comment 7

The authors should improve the discussion comparing the results of their screen with the results obtained by other teams on drug screening on liposarcoma, for example Grad and Coll 2021 PLoS ONE 16(3): e0248140

Response to Comment 7

I really appreciate for your providing the recent drug screening study in liposarcoma. It is noteworthy that HDAC inhibitor demonstrated remarkable antitumor effect in dedifferentiated liposarcoma in both our study and the introduced study. We described it in the revised manuscript.

Round 2

Reviewer 2 Report

I am fully satisfied with the responses provided by the authors 

and the improvements of their revised version which warrants publication in

JPM.